# What promotes or prevents greater use of appropriate compression in people with venous leg ulcers? A qualitative interview study with nurses in the north of England using the Theoretical Domains Framework

Catherine Perry [1], Ross A Atkinson [2], Jane Griffiths,[2] Paul M Wilson [1], Jacqueline F Lavallée,[3] Julie Mullings,[4] Nicky Cullum,[2,5] Jo C Dumville[2]

For numbered affiliations see end of article.

**Correspondence to**
Dr Catherine Perry;
catherine.perry@manchester.ac.uk

## ABSTRACT

**Objectives** To investigate factors that promote and prevent the use of compression therapy in people with venous leg ulcers.

**Design** Qualitative interview study with nurses using the Theoretical Domains Framework (TDF).

**Setting** Three National Health Service Trusts in England.

**Participants** Purposive sample of 15 nurses delivering wound care.

**Results** Nurses described factors which made provision of compression therapy challenging. Organisational barriers (TDF domains environmental context and resources/knowledge, skills/behavioural regulation) included heavy/increasing caseloads; lack of knowledge/ skills and the provision of training; and prescribing issues (variations in bandaging systems/whether nurses could prescribe). Absence of specialist leg ulcer services to refer patients into was perceived as a barrier to providing optimal care by some community-based nurses. Compression use was perceived to be facilitated by clinics for timely initial assessment; continuity of staff and good liaison between vascular/leg ulcer clinics and community teams; clear local policies and care pathways; and opportunities for training such as 'shadowing' in vascular/leg ulcer clinics. Patient engagement barriers (TDF domains goals/beliefs about consequences) focused on getting patients 'on board' with compression, and supporting them in using it. Clear explanations were seen as key in promoting compression use.

**Conclusions** Rising workload pressures present significant challenges to enhancing leg ulcer services. There may be opportunities to develop facilitated approaches to enable community nursing teams to make changes to practice, enhancing quality of patient care. The majority of venous leg ulcers could be managed in the community without referral to specialist community services if issues relating to workloads/skills/training are addressed. Barriers to promoting compression use could also be targeted, for example, through the development of clear patient information leaflets. While the patient engagement barriers may be easier/quicker to address

## STRENGTHS AND LIMITATIONS OF THIS STUDY

⇒ Using qualitative interviews with nurses delivering wound care enabled an in-depth exploration of what promotes or prevents the use of compression therapy in people with venous leg ulcers, from the perspective of staff.

⇒ Initial inductive analysis of interview data ensured that all data were accounted for, and the use of the Theoretical Domains Framework provided a theoretical structure for understanding factors that promoted or prevented compression use.

⇒ Participating nurses were from three Trusts (provider organisations) in the north of England; staff in other geographical regions may have other views.

⇒ All interviews were conducted over the telephone due to the COVID-19 pandemic, so those less happy with using the telephone may not have participated.

than organisational barriers, unless organisational barriers are addressed it seems unlikely that all people who would benefit from compression therapy will receive it.

## INTRODUCTION
### Background
Venous leg ulcers (VLUs) are common, open wounds on the lower leg which take more than 2 weeks to heal.[1] Problems with the veins in the leg (such as damage to the valves or blockages), or impaired action of the calf muscles, prevent optimal blood flow to the heart. This causes venous stasis, increasing the pressure in the veins and triggering a complex change in tissue physiology that can result in open wounds which heal slowly, if at all. Leg ulcers are more prevalent in older people. They can be painful, impair mobility and sleep, restrict work and leisure activities

and reduce psychological well-being, impacting greatly on health-related quality of life.[2–4] Complete wound healing has been identified as the main outcome desired by people with VLUs.[5]

In the UK, VLU care is commonly delivered in the community, in patients' homes or clinics, by nurses or other health professionals. While a wide range of largely nurse-delivered community-focused interventions are available to treat VLUs (bandages, hosiery, dressings and topical agents among others), only compression therapy is supported by a convincing level of evidence[6] and recommended by clinical guidelines.[7 8] Compression therapy, in the form of bandages or stockings, is cost-effective and has become the first-line, evidence-based treatment for VLUs.[6 9] In addition, compression hosiery is current standard practice for prevention.[10]

Compression works when used as intended, but delivery and adherence can be problematic. For example, previous research has highlighted unwarranted variation in clinical practices across wound care services in the north of England.[11] This included significant underuse of compression therapy where clinically appropriate. Understanding how and why suboptimal delivery of evidence-based practices happens is an essential prerequisite to developing strategies to enhance the quality of care delivered. While there is recognition that patient factors may play a role, including discomfort, patients' understanding of the causes of ulceration and their beliefs about compression in terms of healing and symptom control,[12 13] less is known about the influences on professional behaviours. The aim of this study was to explore, from the perspective of nurses delivering wound care, the factors that promote or prevent the use of compression therapy for people with VLUs.

## Theoretical underpinning

Understanding existing professional practices with a view to promoting positive changes in individual and collective behaviour requires an understanding of the influences on behaviours in the context in which they occur.[14] The Theoretical Domains Framework (TDF) is a comprehensive, theory-informed approach for identifying determinants of professional behaviour (cognitive, affective, social and environmental).[11 12] Organised across 14 conceptual domains (see table 1), the TDF has been applied across several healthcare settings to explore professional practices and behaviours.[14 15] We used the TDF to investigate the factors that promote and prevent the use of compression therapy as a treatment for VLUs; thus identifying potentially modifiable determinants of implementation and providing a theoretical foundation to link these to targets for behavioural change.

## METHODS

### Sample

We purposively sampled participants from three National Health Service Trusts in the north of England (Trust 1

**Table 1** The Theoretical Domains Framework: domains and definitions

| Domain | Definition |
|---|---|
| Knowledge | An awareness of the existence of something |
| Skills | An ability or proficiency acquired through practice |
| Social/professional role and identity | A coherent set of behaviours and displayed personal qualities of an individual in a social or work setting |
| Beliefs about capabilities | Acceptance of the truth, reality, or validity about an ability, talent or facility that a person can put to constructive use |
| Optimism | The confidence that things will happen for the best or that desired goals will be attained |
| Beliefs about consequences | Acceptance of the truth, reality or validity about the outcomes of a behaviour in a given situation |
| Reinforcement | Increasing the probability of a response by arranging a dependent relationship or contingency between the response and a given stimulus |
| Intentions | A conscious decision to perform a behaviour or a resolve to act in a certain way |
| Goals | Mental representations of outcomes and states that an individual wants to achieve |
| Memory, attention and decision processes | The ability to retain information, focus selectively on aspects of the environment and choose between two or more alternatives |
| Environmental context and resources | Any circumstance of a person's situation or environment that discourages or encourages the development of skills and abilities, independence, social competence and adaptive behaviour |
| Social influences | Those interpersonal processes that can cause individuals to change their thoughts, feelings or behaviours |
| Emotion | A complex reaction pattern, involving experiential, behavioural and psychological elements, by which the individual attempts to deal with a personally significant matter or event |
| Behavioural regulation | Anything aimed at managing or changing objectively observed or measured actions |

comprised 12 community nursing teams, one community leg ulcer clinic and one secondary care specialist clinic; Trust 2 comprised 6 community nursing teams; Trust 3 comprised 11 community nursing teams). All were qualified nurses providing leg ulcer care within the community, including community nurses and nurses who worked in hospital outpatient clinics.

### Participant recruitment and data generation

Recruitment and data generation took place between February 2020 and January 2021. Potential participants

were contacted by their manager, who provided them with a brief description of the study and a link to an online form to register their interest in participating. Those completing the online form were contacted via email, including a participant information sheet, by a researcher. If they were happy to participate a date and time were arranged for interview. Three potential participants who completed the online form did not go on to interview. Sampling was continued until it was judged that data saturation had been reached.[16]

We had planned to conduct face-to-face interviews, however, because of COVID-19 restrictions, all interviews were conducted over the telephone. Verbal consent was taken and digitally audiorecorded.

We undertook semistructured interviews as they offer a good way to generate data on individuals' experiences.[17] We developed a semistructured interview schedule (online supplemental file 1) with reference to the research aim, the literature and the TDF. Interview questions defined the area to be explored[18] but allowed the interviewer or interviewee to diverge in order to follow up particular areas in more detail.[19] With the permission of participants, interviews were digitally audiorecorded and transcribed.

### Data analysis

We uploaded interview transcripts onto NVivo software to aid data management.[20] Initially, an inductive thematic analysis was undertaken to create a framework for the data.[21] This was followed by a deductive analysis in which the data from the framework were mapped on to the TDF domains.

The initial inductive analysis was carried out by the lead qualitative researcher who had conducted all of the interviews (CP) and to ensure dependability[22] the developing analytical framework was shared and discussed with the research team. Similarly, the deductive analysis mapping specific barriers and enablers within each TDF domain was carried out by one author (CP) in regular discussion with the research team, with additional sense-making and insight provided by the study steering group. To ensure credibility,[22] a near final version of the analysis was presented to a group of 14 nurses experienced in wound care from across England during an online workshop arranged by the National Wound Care Strategy Programme.[8] They were asked about their views on the barriers to compression therapies identified in the study and how best practice may be enabled, and agreed that the findings (and our interpretation of them) reflected their own experiences and/or made sense to them.

### Patient and public involvement

A patient representative sits on the Study Steering Group and has been an active member of the team, contributing to the work from the writing of the study protocol through to discussion of emerging findings and their interpretation. The study was also presented to the University of Manchester Wounds Research Patient and Public Involvement Forum. In particular, members of the forum were consulted about interview schedule questions covering issues pertinent to them, for example, questions regarding the importance of the staff–patient relationship.

## RESULTS

### Participant characteristics

We interviewed 15 nurses who delivered wound care, with interviews between 25 and 69 min in length. Interviewees were all female and comprised nine community nurses or managers, three vascular nurse specialists and three leg ulcer nurses, with a range of experience represented (table 2).

### Relevant domains and key themes

We coded data to seven TDF domains: environmental context and resources, goals, knowledge, skills, beliefs about consequences, behavioural regulation and intentions. The domains of knowledge and skills are presented together as they were linked in the data by the issue of training. We did not have any data that related to the other TDF domains. We did not expect to have data in all of the domains due to the comprehensive nature of the framework. Domains and their associated themes are displayed in table 3. The findings are presented in relation to the TDF domains, with the domains to which most data were coded presented first. Quotations from participants are used to illustrate points made, and are identified with a participant ID and indication of whether they are primarily community based (co) or clinic based (cl).

### Environmental context and resources

#### Lack of commissioned leg ulcer services

In some areas, the lack of specifically commissioned leg ulcer services was mentioned as a barrier to the provision of appropriate compression therapy; in these areas leg ulcer care was provided as part of general community nursing services: 'It's just wrapped up in district nursing.' (014co). Community nurses discussed the challenge of offering optimal care, either because of perceived lack of skills in compression therapy or because of other responsibilities, resulting in apparently eligible people not receiving compression: 'You're not able to follow up….and like offer the treatment as they should anyway because of all the other things you have to worry about.' (002co). Non-housebound patients were not eligible for home visits from community nursing services; in some areas they could attend community nurse-led clinics but in others they received leg ulcer care from practice nurses at their GP surgery, and these nurses were perceived as less likely to be trained in compression therapy: 'Sometimes the practice nurses are not competent to do the compression bandaging.' (004cl). These issues could affect the type of compression a patient was offered.

In areas with commissioned leg ulcer services and leg ulcer clinics, the clinics were highly thought of and seen

**Table 2** Participant characteristics

| Participant ID (cl clinic-based role, co community-based role) | Job title (self-reported) | Length of service as qualified nurse (years) | Time in current role (years) | Highest educational qualification | Training |
|---|---|---|---|---|---|
| 001cl | Sister, community leg ulcer service | 31–40 | 11–20 | Diploma | N18 (compression therapy)<br>V150 (prescribing) |
| 002co | Community specialist practitioner student | 6–10 | 6–10 | Currently completing BSc in District Nursing | 'In house' leg ulcer training |
| 003cl | Vascular leg ulcer nurse | 21–30 | 0–5 | BSc | Tissue viability and leg ulcer management modules (accredited)<br>Ongoing wound management, leg ulcer updates, courses, conferences. |
| 004cl | Vascular nurse specialist | 11–20 | 6–10 | MSc | Leg ulcer management (University, stand-alone module)<br>Wound debridement (University, stand-alone module)<br>Attendance at Wounds UK Conferences |
| 005co | Community staff nurse | 11–20 | 6–10 | Diploma in Nursing with advanced standing | Leg ulcer management Level 6<br>V150 (prescribing)<br>Online wound care training sessions |
| 006co | Community staff nurse | 21–30 | 11–20 | BSc (hons) nursing studies | N18 leg ulcer course<br>Leg ulcer conferences including Lindsay Leg Club<br>Annual 'in-house' wound care training |
| 007cl | Clinical nurse specialist (vascular) | 31–40 | 11–20 | Diploma in nursing | Leg ulcer management<br>Mentorship (supporting learning in practice)<br>Community practitioner nurse prescribing<br>Vasculardisease management(university) |
| 008co | District nursing sister | 6–10 | 6–10 | Specialist practitioner District nursing | Leg ulcer management (university)<br>V150 Community prescriber. |
| 009co | Leg ulcer nurse | 21–30 | 0–5 | RGN | Leg ulcer management (online, university) |
| 010cl | Leg ulcer nurse | 11–20 | 0–5 | Diploma in nursing | Leg ulcer management masters module (online)<br>Wound UK courses<br>Lindsay leg club course |
| 011co | Practice educator for community vascular services | 11–20 | 0–5 | Advanced diploma in adult nursing | Leg ulcer management masters module (online)<br>Nurse prescriber for community formulary<br>Trust 2-day wound care course,<br>Regular TVN and VNS wound care updates |
| 012co | Senior community nurse | 0–5 | 0–5 | BA nursing | Leg ulcer course (1 day) run by Urgo<br>'In house' courses<br>Nurse prescriber |
| 013co | Community nurse | 0–5 | 0–5 | BSc (hons) nursing | Trust training—wound care and compression |
| 014co | Neighbourhood lead | 11–20 | 11–20 | Masters (research) | Numerous training courses over years |
| 015co | Community nurse | 6–10 | 6–10 | RGN | Leg ulcer module (10 days)<br>Drug company study days |

BA, Bachelor of Arts; BSc, Bachelor of Science; MSc, Master of Science; RGN, registered general nurse; TVN, tissue viability nurse; VNS, vascular nurse specialist.

**Table 3** Domains and associated themes

| Domain | Theme |
|---|---|
| Environmental context and resources | Lack of commissioned leg ulcer services<br>Delays in initial leg and wound assessment<br>Lack of continuity of care<br>'Joined up' services and multidisciplinary teams<br>Prescribing issues<br>Application of compression as physically hard work |
| Goals | Getting patients 'on board'<br>Explaining time frames around healing |
| Knowledge and skills | Gaps in knowledge and skills<br>Training |
| Beliefs about consequences | Encouraging compression use<br>Response to non-adherence |
| Behavioural regulation | Variations in bandaging systems used |
| Intentions | Local policies and leg ulcer pathways<br>Compression as treatment of choice |

as facilitating good quality care, incorporating appropriately trained staff and continuity:

> The leg ulcer clinics in [name of area] are fabulous. Fabulous for patients, great for continuity, but also, great for anybody who has joined the district nursing team to make sure we've got that consistency across the [area]. (014co).

Questions were, however, raised about the accessibility of some clinics, and it was suggested that patients may have difficulty getting to clinic locations on multiple/frequent occasions.

### Delays in initial leg and wound assessment

The importance of fast, timely, initial assessment of someone with a suspected VLU was a universal theme: 'Getting the treatment in place quickly, because otherwise it can become….like the chronicity increases and the chance of healing decreases.' (004cl). However, delays in initial assessment were often described. This could be because community nurses did not recognise the need for specialised assessment of a wound: 'Nobody's thought about doing a Doppler. And that might be because they're not experienced, so hadn't thought about it.' (006co); or that they lacked the skills to carry it out: 'There's only a select few of us within the team who can complete the Doppler assessment……there's just not enough of us.' (005co); (the Doppler assessment measures Ankle-Brachial Pressure Index (ABPI) and is an important component of initial assessment prior to compression use). Delays in access to Doppler were also ascribed to the general pressure of work in the community:

> So, pressure of work most definitely is a barrier…… you could go in, see the patient……you know put the dressing on. You are in and out of that house in 15 minutes, aren't you? You can also send your un-registered, or registered but non-nurse members of

the team and so, even though our leg ulcer pathway, if you like, says, the gold standard, after two weeks you should be doing a Doppler, pressure of work, if you've got loads and loads of visits and not many staff and you've got low grades of staff, well then, that can slip can't it? (014co).

Even in areas with dedicated leg ulcer clinics delays in getting an initial assessment were described:

> There needs to be more of them [leg ulcer clinics], because patients are still waiting too long for assessment. So we see lots of patients in vascular [clinic] that have waited three, four months. So they've been seeing the district nurses, waiting three or four months to go and see the leg ulcer clinic, and in that time their leg's getting worse and worse and worse, and in the end, the GP's referred them to Vascular. So they've ended up in secondary care, because the waiting lists in primary care are too long. (003cl).

Two interviewees highlighted the value of specific assessment clinics which had been established to enable timely assessment and one spoke about a new role that she had undertaken providing community vascular support, which she said could lead to more timely assessments. More broadly, interviewees spoke about promoting awareness of leg care. One talked about 'a Legs Matter week…. what they tend to do is promote healthy legs and promote compression and promote good skincare' (006co) and another suggested 'it's not just patients, just nurses in general need [awareness raising]' (001cl).

### Lack of continuity of care

A lack of continuity of nursing care, with different staff attending patients, was perceived as a barrier to effective compression therapy. It was more a problem for community teams than in leg ulcer clinics, and could result in staff not having an overview of a patient's care, knowledge of their disease or treatment history, or reasoning behind clinical decisions. A nurse explained:

> The teams are so large, what will happen is, somebody will start the reduced compression. And then obviously, a different person goes in next time, and they will think they're in reduced compression, because they've got a problem with their arterial line. And no one will ever increase that compression, because they assume, because they're in reduced compression, that they've got some, you know… (003cl).

This situation could be exacerbated by the lack of availability of electronic patient records in the community. A vascular clinic nurse commented:

> So that's another thing that adds to it. When everyone's on different [patient record systems]. 'We can't access the [electronic] hospital records'. I don't think the district nurses have got access. We have but they don't. (010cl).

Lack of continuity in staffing was perceived as leading to a situation where 'somebody else thinks somebody else's, you know, it's somebody else's responsibility' (006co). One interviewee described a change in working practices, 'zonal working' where a smaller group of nurses covered a defined geographical area, as increasing the likelihood of staff continuity. In addition, patients having contact details for the nursing team, for example, through a telephone help-line, was thought to enhance continuity of care and communication between staff and patients.

### 'Joined up' services and multidisciplinary teams

A lack of communication between different services was identified as a barrier to optimal care. A patient with a VLU could be seen by hospital vascular services, leg ulcer clinics and district nursing services, as well as have contact with primary and secondary care services for issues other than their VLU. It was reported that these services did not necessarily communicate well with each other, resulting in unnecessary duplication (eg, where a patient gets referred to both a leg ulcer clinic and a vascular clinic) or unnecessary referral (eg, where a patient had undergone investigations while a hospital inpatient but the results of these investigations were not communicated to the other services). Patients not receiving their compression therapy while hospital inpatients was frequently mentioned:

> Another barrier to our work is that we can be working really hard with a patient, they become unwell, they're admitted to hospital for something completely different. And the first thing they do is take the compression bandages off their legs and just stick a plaster on. Because nobody within hospital can do that treatment either. (001cl).

Conversely, good liaison between vascular or leg ulcer clinics and community nursing teams was described as a facilitator of good care. Often a patient would be seen in a vascular or leg ulcer clinic, a treatment plan drawn up and they would be returned to community teams to continue the treatment. The majority of both clinic-based nurses and community-based nurses reported good communication and liaison between the two: 'Any of the patients now that come to see me leave this clinic with a concise treatment plan……I've got good established links with community' (003cl); 'We have the numbers of all the vascular specialist nurses, so if we feel like we just need to ask a little bit of advice… they'll get back to us straight away' (013co).

Multidisciplinary teams, including nurses, physiotherapists, podiatrists and social workers working well together, were described as facilitating high quality VLU care and compression therapy. The importance of working with the care system was also highlighted in terms of care workers being able to support patients with their compression therapy. One nurse also suggested working with care workers on training:

> The care providers, because they're often quite low-paid, so they have minimal training and that's why we're, our vision is joint training. Because, if they could come on our leg ulcer morning or updates, so that they understand the importance of the compression. (014co).

### Prescribing issues

Compression therapy is a prescribed treatment. Nurses who had been trained as independent prescribers were able to prescribe compression, others had to ask a patient's GP to prescribe the treatment. This could cause delays and could result in patients not having prescribed treatment in time for their next nurse appointment:

> [Patient takes] your form to the GP. It has to sit at the GP's for 48 hours. They then do a prescription. The patient has to pick the prescription up, take it to the chemist. You might wait another 48 hours to pick your prescription up. (001cl).

Most clinic nurses and some community nurses had stocks of compression which they could use if necessary (patients were often started off in their compression from these stocks), but this was not always the case.

### Application of compression as physically hard work

Four interviewees reported that application of compression therapy, in some circumstances, can be physically hard work, particularly where the application takes place in patients' homes without any aids including treatment couches at the correct height. One interviewee commented: 'for district nurses it will often be on the floor on their hands and knees….so it is physically a little bit tricky' (004cl) and another: 'we've had a couple of staff that had got either bad backs or bad knees. So that takes them out of being able to do it for a couple of weeks' (012co). Suggested facilitators were working in pairs, and spreading out compression visits over several days in order to avoid having too many on one day, but these solutions often depended on adequate staffing.

### Goals

#### Getting patients 'on board'

All interviewees described the importance of getting patients 'on board' with their treatment, that is, in agreement with and involved with it, from the beginning: 'I think if they're not on board from the beginning, you're fighting a losing battle.' (005co). This could be difficult: 'Getting them on board to comply with the compression, and the correct compression. I think basically, that's the biggest one that we have a problem with.' (007cl). Explanations were seen as key to engaging patients: nurses were clear that if patients understood that their VLU was a symptom of an underlying long-term condition, and understood the mechanism of compression therapy, they were more likely to adhere to treatment.

In relation to explaining aetiology and compression therapy, interviewees talked about the availability and

utility of written information to use with patients. A minority did not use written information at all and some thought that patients did not read information leaflets anyway. However, the majority of interviewees thought that written information could be useful:

> So it just gives them something to read at their own leisure, and then next time you go, they've got a bit more of an understanding of what you're talking about as well. So I find the leaflets do work quite well. (008co).

Some interviewees had access to 'Trust leaflets' and product company leaflets were also mentioned, although worries were expressed about bias, which may be indicative of variation in the quality of this information. In order to be useful all interviewees thought that leaflets needed to have very simple, clear explanations, some said that pictures were helpful, and one person suggested alternative information formats such as short films.

In terms of getting patients 'on board', the majority of interviewees also spoke of giving patients options or choices, which might make them more likely to adhere to their compression therapy. One nurse explained:

> Like this is how I would feel you'd do best, however, these are your other options. And I'd rather give options at this point and formulate a plan, because you tend to get a little bit more concordance than saying well, this is what you are going to do. (001cl).

There were a number of situations in which it could be possible to offer options, for example, less obtrusive compression or being flexible with clinic appointments. All nurses agreed that it was important to build a relationship with their patient in which they could discuss treatment issues and work together towards the best outcome for the patient: 'A lot of it is getting them on board, building that relationship up with your patients.......and it's a journey together really.' (007cl); 'It's teamwork, it's not a one-way system.' (002co).

### Explaining time frames around healing

The difficulty in estimating how long a VLU may take to heal and explaining this to a patient could be a barrier to setting goals and the successful use of compression therapy. All interviewees said that patients always asked about how long healing would take: 'I think that's one of the first things they'll ask you.' (001cl). Nurses suggested that giving a specific time frame should be avoided: 'The worst thing you could do is give the time. Because, then they're going to sit and wait and say right it's been this long, and it's not healed.' (008co). A number of things that interviewees thought helped in explaining time frames included being positive; talking about healing in small steps; describing what patients can do to speed up healing and issues that may slow healing; and being honest about ulcers that are likely to take a long time to heal, if at all.

One particular difficulty with trying to explain time frames around healing was that in many cases nurses suggested healing of a VLU does not take a straightforward trajectory over time, and a wound may appear to get worse. Interviewees described patients becoming 'really disheartened' (005co). There was consensus that there was usually a discernible cause for an ulcer deteriorating, thus the important thing was to identify the cause and address it: 'You can try and look at a reason why it has happened, like infection. And then treat that cause....... and tell them [patient] that you've addressed that so there is some hope for them that will work.' (004cl).

### Knowledge and skills
#### Gaps in knowledge and skills

There is some suggestion that not all community nurses have the necessary knowledge to recognise the need for a comprehensive leg and wound assessment, or the necessary skills to carry out this initial assessment (including ABPI). Interviewees also recognised a lack of skills among some community nurses in relation to the application of compression bandages: 'The next district nurse that's gone in is not competent in bandaging, so they've not been able to receive the treatment.' (003cl); and the monitoring of compression therapy:

> [Patient] who I sent back to the district nurses to look after......nobody had bothered to put it [compression bandages] back on because she didn't really want it on. So nobody had gone through all the things again, why you need to have it, why it's going to help you....nobody had bothered putting it back on because she'd said 'Oh I don't really want it'. (010cl).

#### Training

Interviewees identified issues with the provision of training to address the gaps in knowledge and skills as being a barrier to the provision of compression therapy. Although training provision was generally considered to be adequate for those working in more specialised roles, for community nurses this was not always the case. A minority of interviewees thought that the availability of training was not good, and a majority mentioned access being more of an issue—it could be challenging to take time out to attend training because of workloads: 'The first thing that gets cancelled is training, unfortunately, 'cos obviously the needs of the patient come before training.' (005co). It was also noted that staff turnover in the community was high, so there was always a need for training. Some positive examples of training were given; a community nurse talked about training offered by her Tissue Viability Team which she described as 'really good....we do it yearly, so we do have leg ulcer training yearly and compression training' (005co). Various facilitators of training were suggested, for example, it was thought that provision of training 'on site' may get around issues of time to attend. Another interesting issue was the kind of 'informal' training that could be gained

by nurses shadowing others in leg ulcer clinics, where these were available: 'You could go and spend an afternoon there and it's just that……The experts are there, great.' (014co).

### Beliefs about consequences
#### Encouraging compression use
All interviewees thought that frequent reiteration of explanations about compression being the best therapy was important in encouraging patients to continue using it. They talked about patients being able to 'self-manage' their care and compression therapy as a goal to be aimed for, which they perceived could contribute to the wider use of compression. However this 'can be difficult sometimes, but we're trying to encourage that self-caring as well and encourage them to take control over their care' (002co). One nurse suggested that barriers to self-care involved not only patients but the thinking and attitudes of nurses themselves:

> Self-management's quite a new…when I say new, I mean sort of, in the last few years. But it was always, sort of, the nurse came and did your dressing, and the nurse looked after it. And, I think, even for staff who've been around a long time it's hard getting that sort of, out of your head. So, it's about, sort of, changing your whole attitude and going in from the start and with the idea [that the patient will self-manage]. (006co).

#### Response to non-adherence
Many interviewees spoke about how their response to a patient who was not adhering to compression therapy was important in terms of whether that person might carry on, or would try compression again in the future: 'We have to ask why and is there anything else we can do to change that or make it better. Try and find the root of the cause of why they're saying no and then take it from there.' (002co). It was also universally agreed that there will always be patients who will not tolerate or adhere to compression therapy, and that this was another point at which there was a need to be sensitive in responding in order to support the patient and keep the door open for them to try compression again in the future:

> I think if they are very sure they don't want to, and we do see…we do see a few like that, you just, I mean, you just kind of say that you understand their choice and, of course, it is their choice what they have on and we understand the reasons why. But that they have to also understand that, you know, healing will be delayed, or indeed will be very difficult to achieve without compression, in a nice way. (004cl).

### Behavioural regulation
#### Variation in available bandaging systems
Bandaging systems available for use varied geographically, as recommended by local Tissue Viability or Leg Ulcer

Teams. When asked whether she was constrained in what compression therapy she could use for a patient based on where a patient lived and the bandage system used there one clinic-based nurse replied: 'I tend to because you just don't want to make life any more difficult' (003cl). Community nurses sometimes changed the compression type decided on by specialist services because they are not able to provide it in their geographical area: 'We change them to [name of compression brand] because that's what we can……that's what our ordering system allows us to order' (012co).

### Intentions
We did not map any barriers to this domain. Instead, the majority of interviewees talked positively about local policies and leg ulcer pathways which they worked within, informed by National Institute for Health and Care Excellence/Scottish Intercollegiate Guidelines Network.[1] [7] These were described as useful by interviewees and they talked about referring to them if they were stuck about where to go next with a patient. They were thought to be particularly useful for less experienced nurses. Interviewees articulated conviction that compression was the treatment of choice for someone with a VLU (despite the possibility that some nurses may not have the skills to put it into practice):

> I think we are very, sort of, strong in the terms that we are very sure…I feel like we, sort of, give quite a lot of confidence to that treatment. Like, we're just very sure that that is the treatment….But I think that helps that we are very clear that we know what the answer is. And the answer is compression. (004cl).

### DISCUSSION
Unwarranted variation in clinical practices across wound care services has been highlighted previously.[11] This study sought to understand how and why suboptimal care happens. In doing so we have used the TDF[14 15] to explore influences on the provision of compression therapy for people with VLUs in the UK. Our analysis suggests that suboptimal compression use is not necessarily due to lack of knowledge of appropriate care pathways, as has been described elsewhere.[23] Instead healthcare staff described a number of factors which they perceived made the provision of compression therapy more challenging. These can broadly be divided into two groups: organisational barriers (TDF domains environmental context and resources, knowledge, skills, behavioural regulation); and patient engagement barriers (TDF domains goals, beliefs about consequences).

### Organisational barriers
Our findings echo Franks et al's[24] suggestion that the workload of community nursing services can impact on whether and how compression therapy is delivered. Rising patient caseloads are presenting growing challenges in

community services,[25] due to both an ageing population and to significant decreases in the number of community nurses.[26] In this study, heavy workloads were seen as contributing to delays in initial leg and wound assessment and in monitoring progress. Work pressures could result in patients being attended by nurses without the necessary knowledge and skills to provide and monitor compression; a decline in the number of qualified district nurses (holders of the specialist practitioner qualification in district nursing) in community nursing services[27] could be a contributory factor. Pressured workloads contributed to a lack of continuity of care too; perceived as problematic by nurses in this study as well as being a concern of patients who were interviewed about their experience of using compression therapy (Perry *et al* 2021, unpublished).

Problems related to the knowledge and skills of healthcare practitioners in leg ulcer care and compression therapy[23 24] and the provision of adequate training[28] have been described. Traditionally, leg ulcer care has been provided through community nursing services. Interviewees in this study generally considered they had the knowledge and skills to manage VLUs themselves, but they identified lack of skills in other community staff, and lack of access to training. The main issue described was the time to attend training, alongside high staff turnover resulting in a constant need, and it was suggested that training provided 'on site' might help. Leg ulcer clinics where community nurses were encouraged to 'shadow' specialist leg ulcer nurses as a form of informal training, which could be fitted around their workload, were valued highly.

In this study, community nursing interviewees perceived the absence of specifically commissioned community-based specialist leg ulcer services, to which they could send patients, as a barrier to providing optimal leg ulcer care. Although referral to a specialist service may be necessary for complex leg ulcers, the majority of ulcers could arguably be managed in general community services.[29] Rather than referring to a specialist service, it may be that if issues relating to delivery models, workloads, skills and training in the community were addressed, appropriate care for patients with VLUs would be enabled. The National Health Service England-funded National Wound Care Strategy Programme is supporting a number of sites in the implementation of its lower limb wound recommendations to explore service delivery models and their potential for success.[8]

Organisational barriers to the provision of compression therapy related to prescribing were described. There were variations between bandaging systems available in different geographical areas which meant that not all compression therapies were available in all areas. Local restrictions on the types of compression systems available have been noted in other studies.[30] This could lead to difficulties for clinic or hospital-based nurses, whose patients came from different geographical areas, as they needed to be aware of different availabilities and prescribe accordingly, leading to a 'postcode lottery' of compression therapies. Including commissioners in discussions about service provision and feeding back about the impact of commissioning decisions may be useful in this situation.

Issues with prescribing were also noted when nurses were not qualified to prescribe compression therapies for their patients, instead having to request a prescription from a patient's general practitioner. This could lead to delays in patients starting compression therapy or a temporary break in their treatment regimen and interviews with patients found evidence of patients not being in compression because they were waiting for prescribed hosiery to arrive (Perry *et al* 2021, unpublished).

### Patient engagement barriers

A good relationship between patient and health professional is an important foundation for the successful use of compression therapy.[31–33] In our study, nurses spoke of the importance of getting patients 'on board' with their compression therapy, and of supporting them in using it. Explanations were seen as key and it was almost universally thought that a patient who understood the aetiology of their VLU and the mechanism of compression therapy was more likely to continue to adhere to treatment. However, it was evident in a study of patients (Perry *et al* 2021, unpublished) that there was variation in patients' perceived understanding of the cause of their VLU and that many people had difficulty understanding this topic. There was not a clear relationship between knowledge/understanding and compression use: individuals who did and did not have an understanding of VLU aetiology or how compression worked, were adherent. It is possible that for some individuals the important thing was that explanations were offered: they were happy to rely on the recommendation of their nurse. This highlights the importance of the patient/health professional relationship.

Keeping patients 'on board' with their treatment was made harder because of the difficulties in explaining how long a VLU may take to heal and in supporting patients if their VLU become worse rather than improving. Patient interviews (Perry *et al*, 2021, unpublished) confirm that patients wanted to know how long healing might take and became very disheartened if their VLU became worse, thus underlining the importance of these issues being handled carefully by nurses. Encouraging some degree of 'self-management' or perhaps more accurately 'self-care' was thought to encourage continued compression use, although there is limited evidence of the benefits or risks of this in relation to compression therapy.

### Intervention for change

More data were coded to organisational barriers in this study (TDF domains environmental context and resources, knowledge, skills, behavioural regulation) than to patient engagement barriers (TDF domains goals, beliefs about consequences). This could suggest that a multicomponent intervention will be required to support

further delivery of compression in the treatment of VLUs. We recognise that rising patient caseloads and related workload pressures will present significant challenges to any attempt to enhance existing leg ulcer services. Regardless, the organisational barriers highlighted in this study are similar to those highlighted by previous research,[11] but crucially this also suggests that there are similar opportunities to develop facilitated approaches to enable community nursing teams to make changes to practice and enhance the quality of patient care delivered.[25] This information can therefore be used to inform future improvement efforts locally.

Our analysis also suggests that there may be traction in addressing the barriers highlighted in relation to promoting appropriate compression use in people with VLUs. To this end, we have been working with one of the healthcare provider organisations involved in this work to produce an information leaflet which can be used by all staff to explain and promote the use of compression therapy. Strategies to encourage 'self-management' or 'self-care' would also merit further investigation.

### Study limitations

Although staff from the three participating community services were included in the sample, the majority came from one service. However, the issues raised by staff from all three services were similar. The research was conducted during the COVID-19 pandemic. All participants were therefore interviewed over the telephone. This may mean that our study participants were those who were naturally more comfortable with and experienced in talking over the telephone.

There were some challenges in using the TDF, as reported by other users.[8] Some data could be coded to different domains, decisions about coding in these instances were made through discussion between the research team. More data were coded into the environmental context and resources domain than any other. As anything that a healthcare professional does could be viewed as dependent on context,[14] the coding to this domain was examined carefully to ensure that it was the best fit. In one instance coding to the TDF domains split up data (on prescribing) that could usefully be considered together. However, this could be put back together in considering the implications of the findings.

### CONCLUSIONS

We have presented an exploration of the factors that prevent or promote the appropriate use of compression therapy for people with VLUs, from the perspective of nurses involved in their care. While patient engagement barriers may be easier and quicker to address than some of the organisational barriers identified, unless organisational barriers are also attended to it would seem unlikely that all people who would benefit from compression therapy will receive it.

**Author affiliations**
¹Division of Population Health, Health Services Research and Primary Care, School of Health Sciences, University of Manchester, Manchester, UK
²Division of Nursing, Midwifery and Social Work, School of Health Sciences, University of Manchester, Manchester, UK
³Division of Medical Education, School of Medical Sciences, University of Manchester, Manchester, UK
⁴Northenden Health Centre, Manchester University NHS Foundation Trust, Manchester, UK
⁵MAHSC, Manchester University NHS Foundation Trust, Manchester, UK

**Acknowledgements** The authors would like to express thanks to Cheryl Lenney (Chief Nurse, Manchester University NHS Foundation Trust), and Alex Barker (Head of Nursing, Manchester Local Care Organisation) who were members of the Project Management Group (PMG). Thank you to our local Principal Investigators, Louise O'Connor (Lead Nurse for Tissue Viability and Infection Prevention, Manchester University NHS Foundation Trust, who was also a member of the PMG), Joanne Taylor (Tissue Viability Service Lead, Rochdale Care Organisation) and Martin Sylvester (Research Nurse, Mid Yorkshire Hospitals NHS Trust). We also thank our patient representative on the PMG, who has contributed to the work from the outset, including development of the protocol through to interpretation of the findings. The University of Manchester Wounds Research Patient and Public Involvement (PPI) Forum made useful comments on the interview schedule. We thank Francisco Espinoza for setting up the online participant registration form, and the nurse managers at participating Trusts who facilitated recruitment of their staff members as participants. Finally we would like to thank all of the staff who so generously gave their time to participate in this research during a very difficult period for NHS services.

**Contributors** NC conceived the idea and design for the overall project. RAA, JCD, JG, JL and PW contributed to further development of the study design. CP collected the data. CP, RAA, JG and PW conducted the data analysis. CP created the original draft of the manuscript. All authors (RAA, NC, JCD, JG, JL, JM, CP and PW) contributed to the interpretation of study findings, critical revision of the manuscript for important intellectual content and approval of the final manuscript. JCD will act as guarantor for the study.

**Funding** This work was supported by National Institute for Health Research (NIHR) Research for Patient Benefit (RfPB) scheme, grant number PB-PG-0817-20002. NC, JCD, PW and CP are partially funded by the National Institute for Health and Care Research Applied Research Collaboration Greater Manchester.

**Disclaimer** The views expressed in this article are those of the authors and not necessarily those of the NHS, National Institute for Health and Care Research, or the Department of Health and Social Care.

**Competing interests** None declared.

**Patient and public involvement** Patients and/or the public were involved in the design, or conduct, or reporting, or dissemination plans of this research. Refer to the Methods section for further details.

**Patient consent for publication** Not applicable.

**Ethics approval** Ethical approval was received in 2019 from the London Bridge NHS Research Ethics Committee, reference number 19/LO/0826.

**Provenance and peer review** Not commissioned; externally peer reviewed.

**Data availability statement** Data are available on reasonable request. Requests for access to data should be addressed to the corresponding author. Data underlying the study findings will be available on reasonable request. Due to the nature of the ethical approval given for the study, it may not be possible to share complete interview transcripts, but additional anonymised illustrative quotations may be available.

**ORCID iDs**
Catherine Perry http://orcid.org/0000-0002-8496-6923
Ross A Atkinson http://orcid.org/0000-0001-8976-2754
Paul M Wilson http://orcid.org/0000-0002-2657-5780

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
