## [Reviewer comments · BMJ Open]

ARTICLE DETAILS

TITLE (PROVISIONAL)	What promotes or prevents greater use of appropriate compression in people with venous leg ulcers? A qualitative interview study with nurses in the north of England using the Theoretical Domains Framework
AUTHORS	Perry, Catherine; Atkinson, Ross; Griffiths, Jane; Wilson, Paul; Lavallée, Jacqueline; Mullings, Julie; Cullum, Nicky; Dumville, Jo C.

VERSION 1 – REVIEW

REVIEWER	Abu Bakar Aloweni, Fazila Singapore General Hospital, Nursing Division
REVIEW RETURNED	15-Mar-2022

GENERAL COMMENTS	Kudos to the team for carrying such meaningful and important work during this challenging times. Just a few comments and suggestions: 1. revise the findings in the abstract, include the themes to show the link to the domains of the TDF - like how you wrote under discussion.2. how many 'potential' participant was contacted by the manager? just curious how many of the nurses were willing to participate- this is something that could be discussed on.3. Also since they are 'approached' by their manager- is there a way to prevent undue influence or pressure to participate in the study?
---

REVIEWER	Poku, Edith The University of Sheffield, School of Health and Related Research
REVIEW RETURNED	18-Mar-2022

GENERAL COMMENTS	Manuscript - BMJ Open What promotes or prevents greater use of appropriate compression in people with venous leg ulcers? A qualitative interview study with nurses using the Theoretical Domains Framework Thank you for the opportunity to review this manuscript. General comments Questions have been raised about the best approach for compression therapy in patients with venous leg ulcers. Therefore, this study is relevant to those who work in this area as well as to those who receive treatment.
---

The authors demonstrated good knowledge about the topic area and study methodology (in the Introduction, Methods and Discussion) and have presented their work in a coherent manner. However, the manuscript could be further improved by using a more academic style of writing.

additional comments

Section	Reviewer's Feedback
ABSTRACT	Based on the objectives of the study, it will be informative to highlight or specify more clearly the identified factors (barriers and facilitators) that promote and prevent the utilisation of compression therapy. The current subsection tends to discuss the challenges and how these could be addressed (page 3).
FINDINGS	(1) Concerns about the risk of identifying interviewees Due to the sample size and restricted region for sampling participants, details provided in Table 2, pages 9 to 10 could raise concerns about anonymity of participants. For this reason, the authors are kindly advised to revise identifying quotes using both the participant ID and job title (page 15). (2) Improving the context of findings To provide a better perspective in relation to their findings,  • the authors are encouraged to use descriptors that relate to respondents in the community and hospital settings (or relevant settings) rather than the participant ID. • the authors must include the number of participants where appropriate. E.g. Some interviewees (n= 'x' community-based nurses; 'y'= clinic-based nurses) reported that application of compression therapy, in some circumstances, can be physically hard work...(page 15) *Full terms could be abbreviated after first use. Relevant domains and key themes Kindly justify the selection of seven domains
ABBREVIATIONS	Please define an abbreviation in full when used for the first time (NHSE-) page 20. Please include abbreviations with full terms at the end of Tables (e.g. TVN, VNS, RGN in Table 2; Table on pages 29 to 32)

VERSION 1 – AUTHOR RESPONSE

Reviewer 1 comments

Reviewer 1 comment	Response
1 Kudos to the team for carrying such meaningful and important work during this challenging times.	Thank you very much.
2 Revise the findings in the abstract, include the themes to show the link to the domains of the TDF - like how you wrote under discussion.	We have revised the findings section in the abstract (now called 'results' as per BMJ Open policy) to include the themes to show the link to the TDF domains (p.3 in resubmitted manuscript).
3 How many 'potential' participant was contacted by the manager? Just curious how many of the nurses were willing to participate- this is something that could be discussed on.	Unfortunately we do not know how many people were contacted by the managers. They were asked to distribute the invite to everybody in their teams. We do know that three potential participants who completed the online form did not go on to interview. This information has been added to the 'Participant recruitment and data generation' section on p.7 of the resubmitted manuscript.
4 Also since they are 'approached' by their manager- is there a way to prevent undue influence or pressure to participate in the study?	We agree that it was very important to avoid participants feeling pressured to participate. We liaised with the nurse managers to ensure that they used a standardised text to approach their staff, this text was approved by the NHS Ethics Committee which gave ethical approval for the research. The nurse managers sent out the invite only once and did not follow it up, and would not know who did or did not respond to the invitation. These details have been added to the supplementary information on the COREQ form.

Reviewer 2 comments

Reviewer 2 comment	Response
1 Questions have been raised about the best approach for compression therapy in patients with venous leg ulcers. Therefore, this study	Thank you for this comment.

is relevant to those who work in this area as well as to those who receive treatment.	
2 The authors demonstrated good knowledge about the topic area and study methodology (in the Introduction, Methods and Discussion) and have presented their work in a coherent manner. However, the manuscript could be further improved by using a more academic style of writing.	Thank you for these comments. The manuscript has been written following the consolidated criteria for reporting qualitative research (COREQ) guidelines and we have responded to individual comments from the editor and reviewers to produce a document which we consider to be written in an appropriate style.
Abstract 3 Based on the objectives of the study, it will be informative to highlight or specify more clearly the identified factors (barriers and facilitators) that promote and prevent the utilisation of compression therapy. The current subsection tends to discuss the challenges and how these could be addressed (page 3).	We have revised the findings section in the abstract (now called 'results' as per BMJ Open policy) and have specified barriers and facilitators that promote and prevent the utilisation of compression therapy. We have also included reference to the TDF domains as suggested by Reviewer 1. In the conclusion section we have then moved on to discuss how the challenges could be addressed (p. 3 in resubmitted manuscript).
Findings 4 (1) Concerns about the risk of identifying interviewees: Due to the sample size and restricted region for sampling participants, details provided in Table 2, pages 9 to 10 could raise concerns about anonymity of participants. For this reason, the authors are kindly advised to revise identifying quotes using both the participant ID and job title (page 15).	Thank you for this comment. We completely agree that the anonymity of our participants is of utmost importance. In order to ensure this, we have revised the details provided in Table 2 in the following ways. We have removed the date of the highest qualification where it was present, and presented both 'length of service' and 'time in current role' as a range of years rather than an exact figure, as suggested by the journal editor. We have also removed the 'service code' column (Table 2 p. 9-10 in resubmitted document). We have added an indication as to whether a participant is predominantly community-based or predominately clinic-based (as Reviewer 2, comment 5 (2), below). Table 2 has been particularly examined in relation to maintaining the anonymity of individual participants by a paper author who is a practitioner in the field in the north of England, and we are confident that individuals are not identifiable.

	Quotations are labelled with a participant ID throughout the findings, in line with COREQ guidelines. Job titles are not used to identify quotations, although it has now been indicated if a participant was community-based or clinic-based (see comment in Findings 5 (2) below.
Findings 5 (2) Improving the context of findings: To provide a better perspective in relation to their findings, the authors are encouraged to use descriptors that relate to respondents in the community and hospital settings (or relevant settings) rather than the participant ID.	Thank you for this suggestion, we think that it will improve the context of the findings if it is indicated whether participants are predominantly community-based or clinic-based. A comment to this effect has been added on page 11 in the resubmitted document: ‘Quotations from participants are used to illustrate points made, and are identified with a participant ID and indication of whether they are primarily community-based (co) or clinic-based (cl)’. This style of labelling has been applied throughout the results section (pp. 8-19 in resubmitted manuscript). We have continued to use the participant ID as well, as this is in line with COREQ guidelines and helps to demonstrate that a range of participants are represented in the results. We are confident that this way of labelling quotations/participants does not compromise their anonymity (see comment above in Findings 4 (1).
Findings 6 To provide a better perspective in relation to their findings: The authors must include the number of participants where appropriate. E.g. Some interviewees (n= ‘x’ community-based nurses; ‘y’= clinic-based nurses) reported that application of compression therapy, in some circumstances, can be physically hard work...(page 15). *Full terms could be abbreviated after first use.	Thank you for this suggestion, we agree that it can be useful to know whether a particular perspective was held by all interviewees or just one or two. We have gone through the results and have added numbers in some places where it seems appropriate. However we have also often used terms such as ‘the majority’ or ‘a universal theme’ to indicate the frequency of a particular view, in keeping with the purposive (non-probability) basis of our sampling of participants (highlighted in red in the results section in resubmitted paper, pp. 8-19).
Relevant domains and key themes 7 Kindly justify the selection of seven domains.	Having carried out an inductive analysis of the data, we then used deductive coding to code our data to the TDF domains. We coded data to seven of the TDF domains, but we did not have any data that mapped on to the

	remaining seven domains (we did not expect that we would have data in all of the domains due to the comprehensive nature of the framework). We have added two sentences to clarify this under 'Relevant domains and key themes' on page 11 of the resubmitted paper.
Abbreviations 8 Please define an abbreviation in full when used for the first time (NHSE-).	We have defined the identified abbreviation in full (p.21 of resubmitted document) and checked the whole document to ensure that there are no other abbreviations which need defining on first mention.
Abbreviations 9 Please include abbreviations with full terms at the end of Tables (e.g. TVN, VNS, RGN in Table 2; Table on pages 29 to 32).	We have checked that abbreviations with full terms have been listed under relevant tables (Table 2 p 9-10). We have also checked that abbreviations in the interview schedule (Supplementary File 1, which appeared on pages 29-32 of the original collated document) appear in full at first mention.

VERSION 2 – REVIEW

REVIEWER	Abu Bakar Aloweni, Fazila Singapore General Hospital, Nursing Division
REVIEW RETURNED	23-May-2022

GENERAL COMMENTS	Kudos to the team for conducting this meaningful work. Indeed VLU is a complex condition to treat. The authors should consider developing and evaluating strategies to overcome the barriers identified in this study. Here are a few suggestions for improvement:  1. Pg 8, Line 7, For the sake of international reader, perhaps authors could specify and provide context how big the three NHS Trusts is, i.e. how many nurses are there, how many were contacted and responded-so this would also provide readers on the challenges recruiting community nurses for research. 2. Pg 9, line 5. Could you explain how many nurses were involved in reviewing the final version of the analysis and how was it conducted given that these wound care nurses are across England. What questions were asked?
---

REVIEWER	Poku, Edith The University of Sheffield, School of Health and Related Research
REVIEW RETURNED	06-Jun-2022

GENERAL COMMENTS	Dear Author Thank you for resubmitting your manuscript and accompanying responses to previous feedback from reviewers. Reviewer's comments relating to few minor revisions  ABSTRACT Please add (TDF) at the end of the statement as follows: Design Qualitative interview study with nurses using the Theoretical Domains Framework (TDF).  Results and Conclusions The authors have provided useful information here. However, minor revision of language would be helpful to improve clarity and conciseness of the abstract.  Table 2  • Most nursing qualifications are awarded under Health and Medical Science Faculties or institutions. Could the authors kindly confirm the indicated degree, BA Nursing (for 012co) is the right qualification, or should this be BN (Bachelor of Nursing)? • Suggested formatting of abbreviations (as indicated): BA, Bachelor of Arts; BSc, Bachelor of Science; MSc, Master of Science; RGN Registered General Nurse; TVN, Tissue Viability Nurse; VNS, Vascular Nurse Specialist  Consistency throughout the manuscript is encouraged in terms of participant ID and setting of work. The authors note that 'Quotations from participants are used to illustrate points made and are identified with a participant ID and indication of whether they are primarily community-based (co) or clinic-based (cl) [page 11]. Much of the in-text information include a prefix 'S' (e.g., S004cl) but participant ID in Table 2 has no prefix.  Suggestion: Kindly delete parenthesis in the statement and add additional citation(s) if this statement related to other studies. Organisational barriers (page 20) Our findings echo those of other studies (e.g.²⁴) which have suggested that the workload of community nursing services can impact on whether and how compression therapy is delivered.  REFERENCES A few abbreviations need to be presented in full – please see references 1; 7; 27; 28 and 29.
---

VERSION 2 – AUTHOR RESPONSE

Response to reviewer comments

Reviewer 1 comments	Response
Kudos to the team for conducting this meaningful work. Indeed VLU is a complex condition to treat. The authors should consider developing and evaluating strategies to overcome the barriers identified in this study.	Thank you, we agree fully and are working with health care professionals both locally and through the National Wound Care Strategy Programme. One example is the development of patient facing written material, informed by the research and further developed with stakeholder involvement, which is about to be used in practice.
Pg 8, Line 7, For the sake of international reader, perhaps authors could specify and provide context how big the three NHS Trusts is, i.e. how many nurses are there, how many were contacted and responded-so this would also provide readers on the challenges recruiting community nurses for research.	We agree that recruiting community nurses for research can be challenging. We have added some detail about the participating Trusts in order to give some idea of the size of the teams approached: 'We purposively sampled participants from three NHS Trusts in the North of England (Trust 1 comprised 12 community nursing teams, one community leg ulcer clinic and one secondary care specialist clinic; Trust 2 six community nursing teams; Trust 3 11 community nursing teams).' (Page 7 of resubmitted document). Unfortunately we do not know how many people were contacted by their managers with an invitation to participate. Managers were asked to distribute the invite to everybody in the relevant teams, but we do not know how many nurses received this invite.
Pg 9, line 5. Could you explain how many nurses were involved in reviewing the final version of the analysis and how was it conducted given that these wound care	We have added some detail as suggested, which now reads:

nurses are across England. What questions were asked?	'To ensure credibility²² a near final version of the analysis was presented to a group of 14 nurses experienced in wound care from across England during an online workshop arranged by the National Wound Care Strategy Programme.⁸ They were asked about their views on the barriers to compression therapies identified in the study and how best practice may be enabled, and agreed that the findings (and our interpretation of them) reflected their own experiences and/or made sense to them.' (Page 8 of resubmitted document).
Reviewer 2 comments	Response
Abstract Please add (TDF) at the end of the statement as follows: Design Qualitative interview study with nurses using the Theoretical Domains Framework (TDF).	This has been added as suggested (p.3 of resubmitted document).
Abstract Results and Conclusions The authors have provided useful information here. However, minor revision of language would be helpful to improve clarity and conciseness of the abstract.	Thank you. We have made some minor revisions in the results section (p. 3 in resubmitted document).
Table 2 Most nursing qualifications are awarded under Health and Medical Science Faculties or institutions. Could the authors kindly confirm the indicated degree, BA Nursing (for 012co) is the right qualification, or should this be BN (Bachelor of Nursing)?	We have checked this and BA Nursing was the response given by the interviewee when they were asked what their highest educational qualification was.

Table 2 Suggested formatting of abbreviations (as indicated): BA, Bachelor of Arts; BSc, Bachelor of Science; MSc, Master of Science; RGN Registered General Nurse; TVN, Tissue Viability Nurse; VNS, Vascular Nurse Specialist.	We have formatted the abbreviations as suggested (p.10 of resubmitted document).
Consistency throughout the manuscript is encouraged in terms of participant ID and setting of work. The authors note that 'Quotations from participants are used to illustrate points made and are identified with a participant ID and indication of whether they are primarily community-based (co) or clinic-based (cl) [page 11]. Much of the in-text information include a prefix 'S' (e.g., S004cl) but participant ID in Table 2 has no prefix.	Thank you for pointing out this inconsistency. The 'S' prefix on the participant ID is not necessary and has been removed from the text throughout the results section (pp. 8 - 19 in the resubmitted document). Participant ID labels in the text now match those presented in Table 2.
Kindly delete parenthesis in the statement and add additional citation(s) if this statement related to other studies. Organisational barriers (page 20) Our findings echo those of other studies (e.g.24) which have suggested that the workload of community nursing services can impact on whether and how compression therapy is delivered.	We have changed the text to read 'Our findings echo Franks et al.'s²⁴ suggestion that the workload of community nursing services can impact on whether and how compression therapy is delivered' (p.20 of resubmitted document).
REFERENCES A few abbreviations need to be presented in full – please see references 1; 7; 27; 28 and 29.	The abbreviations used in references 1; 7; 27; 28 and 29 have all been presented in full (pp. 24 – 27 of resubmitted document).

VERSION 3 – REVIEW

REVIEWER	Abu Bakar Aloweni, Fazila Singapore General Hospital, Nursing Division
REVIEW RETURNED	12-Jul-2022
GENERAL COMMENTS	The authors have adequately addressed all the questions raised. No further input. All the best to the research team in the knowledge translation